# No Clustering, No Routing: How Transformers Actually Process Rare Tokens

## Abstract

Large language models struggle with rare token prediction, yet the mechanisms driving their specialization remain unclear. Prior work identified specialized "plateau" neurons for rare tokens following distinctive three-regime influence patterns [7], but their functional organization is unknown. We investigate this through neuron influence analyses, graph-based clustering, and attention head ablations in GPT-2 XL and Pythia models. Our findings show that: (1) rare token processing requires additional plateau neurons beyond the power-law regime sufficient for common tokens, forming dual computational regimes; (2) plateau neurons are spatially distributed rather than forming modular clusters; and (3) attention mechanisms exhibit no preferential routing to specialists. These results demonstrate that rare token specialization arises through distributed, training-driven differentiation rather than architectural modularity, preserving context-sensitive flexibility while achieving adaptive capacity allocation.

## 1 Introduction

Large language models (LLMs) achieve remarkable performance across diverse tasks, yet their internal mechanisms for processing different token types remain poorly understood. Rare tokens—those appearing infrequently in training data pose particular challenges: they often encode critical semantic information while exhibiting systematically lower prediction accuracy [5]. Prior studies show that model performance on factual tasks correlates with entity frequency [1] and that truncating rare tokens in training can lead to model collapse [12], underscoring the importance of understanding how LLMs process rare events.

Recent advances in mechanistic interpretability have revealed specialized circuits for specific linguistic computations, including feed-forward layers acting as key-value memories [4] and sparse, monosemantic representations [3]. In particular, Liu et al. [7] found that rare token processing exhibits a distinctive three-regime influence distribution: a plateau of highly influential neurons, followed by a power-law decay and a rapid decay and they argue that this might be related with neuron coordinated in both activation and weight space.

However, while specialized plateau neurons exist for rare tokens, **it is unclear how transformers organize and access these computational resources**. This organizational question connects to complementary learning systems theory [9, 6], where two frameworks make opposing predictions. The **modular hypothesis** suggests specialization requires discrete neuron clusters and selective routing [11], while the **distributed hypothesis** proposes that specialization emerges from parameter-level differentiation within shared substrates [8].

To test these hypotheses, we investigate three questions: (1) whether rare and common tokens require different computational regimes, (2) whether plateau neurons form spatial clusters, and (3)

whether attention mechanisms selectively route rare tokens to specialists. Our results support the distributed hypothesis: plateau neurons provide additional processing capacity while remaining spatially distributed and accessed through universal attention patterns. This challenges architectural modularity assumptions and demonstrates that sophisticated computational organization can emerge from simple training dynamics.

## 2   Methods

We investigated rare token specialization in GPT-2 XL and Pythia models through three complementary analyses: neuron influence, spatial organization, and attention routing. All analyses focus on the final MLP layer, where prior work observed the characteristic three-regime influence pattern [7].

**Dataset and Token Selection**   We sampled 25,088 tokens from the C4 corpus [10]. Tokens were split into rare and common groups based on frequency, using the 50th percentile as a threshold to ensure balanced sample sizes while capturing long-tail effects.

**Neuron Influence Analysis**   To quantify individual neuron contributions, we performed mean ablation experiments. Influence of neuron $n$ was measured as the absolute change in loss after ablation:

$$\text{Influence}(n) = |\mathcal{L}_{\text{ablated}}(n) - \mathcal{L}_{\text{baseline}}|. \tag{1}$$

Neurons were ranked by influence, and a power-law curve was fitted to the distribution. Neurons whose influence significantly exceeded the fit were classified as the plateau regime, while mid- and low-influence neurons followed the standard decay. Comparing rare and common tokens allowed us to assess whether distinct computational regimes emerge.

**Spatial Organization Analysis**   To test whether plateau neurons form modular clusters, we constructed correlation networks from neuron activations and applied Louvain community detection [2]. Signed modularity $Q$ quantifies clustering:

$$Q = \frac{1}{2m} \sum_{ij} \left[ A_{ij} - \frac{k_i k_j}{2m} \right] \delta(c_i, c_j), \tag{2}$$

where $A_{ij}$ is the correlation-based edge weight, $k_i$ the node degree, and $\delta(c_i, c_j)$ indicates same-community membership. Plateau neuron modularity was compared to random baselines to evaluate significance.

**Attention Routing Analysis**   To examine whether rare tokens rely on specialized attention routing, we analyzed attention patterns in layers preceding the final MLP. Attention concentration was quantified via Gini coefficients, and correlations between rare and common token attention distributions were computed. Head-specific contributions were measured using ablation:

$$\text{Impact}(h) = \frac{|\text{Activation}_{\text{baseline}} - \text{Activation}_{\text{ablated } h}|}{\text{Activation}_{\text{baseline}}}. \tag{3}$$

Minimal impact from single-head ablations, compared to large drops from full-layer ablation, indicates that plateau neurons integrate signals from multiple heads rather than being selectively targeted. All analyses included statistical controls and significance testing to ensure that observed patterns reflect genuine specialization rather than noise or sampling variability.

## 3   Results

**Rare and Common Tokens Show Distinct Influence Patterns**   Figure 1 shows that rare and common tokens engage distinct computational regimes. For common tokens, neuron influence follows a well-fitted power-law:

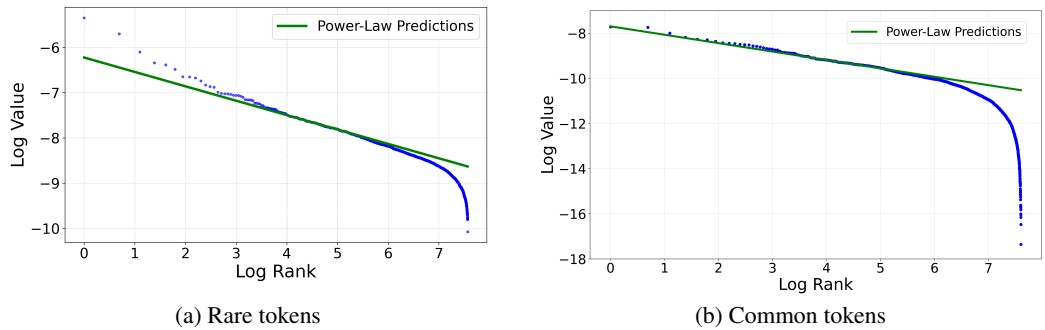

| (a) Rare tokens | (b) Common tokens |

Figure 1: Neuron influence distributions for rare vs. common tokens. Rare tokens exhibit a plateau of specialist neurons, a power-law region, and a rapid decay region. Common tokens follow a pure power-law distribution without a plateau regime.

$$\log |\Delta \mathcal{L}| \approx -\kappa \log(\text{rank}) + \beta, \tag{4}$$

with $\kappa = 1.84 \pm 0.12$ ($R^2 = 0.94$). Deviations from this fit remain small ($|\delta| < 0.1$ for 95% of neurons), confirming that common tokens primarily rely on distributed scaling.

In contrast, rare tokens systematically deviate from power-law behavior. Among the top 15–20 neurons, we observe a clear plateau regime with positive deviations ($\delta > 0.5$) relative to the fitted curve. Beyond the plateau, mid-ranked neurons follow a similar power-law decay, while low-influence neurons show rapid signal attenuation. This establishes a dual-regime structure for rare tokens, consistent across both GPT-2 XL and Pythia families.

**Plateau Neurons Are Spatially Distributed**    We next tested whether plateau neurons form spatial clusters or are distributed across the MLP layer. Using graph-based Louvain community detection, we computed signed modularity scores for excitatory and inhibitory plateau neurons and compared them to random controls (Table 1).

Across both models, all plateau neuron groups had modularity scores comparable to random baselines (Pythia-410M: 0.03–0.05 vs. control 0.04; GPT-2 XL: 0.07–0.11 vs. control 0.09), with no statistically significant clustering (p-values 0.42–0.84). Spectral clustering produced consistent null results ($Q_{\text{plateau}} = 0.08 \pm 0.05$ vs. $Q_{\text{control}} = 0.07 \pm 0.06$, $p = 0.41$, Mann–Whitney $U$ test). Together, these analyses indicate that plateau neurons are spatially distributed rather than forming discrete modules.

Table 1: Community detection results show no significant clustering of plateau neurons compared to random baselines across model scales.

| Model | Neuron Group | Signed Modularity | Communities | p-value |
|---|---|---|---|---|
| Pythia-410M | Plateau (Excitatory) | $0.05 \pm 0.04$ | $1.8 \pm 0.5$ | 0.67 |
| | Plateau (Inhibitory) | $0.03 \pm 0.05$ | $1.9 \pm 0.6$ | 0.84 |
| | Random Control | $0.04 \pm 0.04$ | $1.9 \pm 0.4$ | – |
| GPT-2 XL | Plateau (Excitatory) | $0.11 \pm 0.06$ | $2.3 \pm 0.7$ | 0.42 |
| | Plateau (Inhibitory) | $0.07 \pm 0.05$ | $2.1 \pm 0.5$ | 0.73 |
| | Random Control | $0.09 \pm 0.05$ | $2.2 \pm 0.6$ | – |

**No Evidence for Selective Attention Routing**    To examine whether rare tokens rely on specialized attention routing, we analyzed attention distributions in layers preceding the final MLP. Correlations between rare and common tokens were high ($r = 0.89 \pm 0.07$), and attention concentration (Gini coefficient) did not differ significantly ($G_{\text{rare}} = 0.34 \pm 0.05$ vs. $G_{\text{common}} = 0.32 \pm 0.04$, $p = 0.43$, t-test).

Systematic head ablation experiments further confirmed distributed access (Table 2). Single-head ablations caused small, statistically similar reductions in plateau activation (effect sizes 0.26–0.31),

whereas removing all heads in a layer produced large drops (-42% to -45%). These results indicate that plateau neurons integrate signals from multiple heads rather than relying on dedicated routing circuits.

Overall, our results show that rare tokens (1) recruit plateau neurons not activated by common tokens, establishing distinct computational regimes; (2) engage neurons that are spatially distributed rather than clustered; and (3) access these neurons via distributed attention mechanisms with no selective routing. This supports a general principle of distributed specialization in transformers rather than modular organization.

Table 2: Attention head ablation shows distributed dependency patterns across model scales. Individual heads show minimal impact on plateau neuron activation.

| Model | Ablation Target | Plateau Activation Change | Effect Size | p-value |
|---|---|---|---|---|
| Pythia-410M | Single Head (max impact) | -7.8% ± 2.3% | 0.29 | 0.03 |
| | Random Head (baseline) | -7.1% ± 2.9% | 0.26 | 0.04 |
| | All Heads | -42.1% ± 5.2% | 1.74 | <0.001 |
| | Control (non-attention) | -1.3% ± 1.6% | 0.05 | 0.71 |
| GPT-2-XL | Single Head (max impact) | -8.2% ± 2.1% | 0.31 | 0.02 |
| | Random Head (baseline) | -7.4% ± 3.2% | 0.28 | 0.03 |
| | All Heads | -45.3% ± 4.7% | 1.82 | <0.001 |
| | Control (non-attention) | -1.1% ± 1.8% | 0.04 | 0.67 |

## 4 Discussion and Limitations

Our results demonstrate that transformers handle rare tokens through distributed, training-driven specialization rather than modular architectural design. Rare tokens recruit additional high-influence plateau neurons that are largely inactive during common token processing, establishing dual computational regimes. These plateau neurons are spatially distributed across the MLP layer and are accessed through the same attention patterns used by all tokens, with no evidence of selective routing. Together, these findings indicate that transformers achieve rare token specialization by differentiating parameters within shared computational substrates, preserving flexible and context-sensitive processing without requiring discrete modules.

This distributed organization provides insight into the scalability and robustness of transformer architectures. By leveraging parameter-level specialization instead of hardwired modular structures, transformers can allocate computational capacity adaptively, avoiding bottlenecks while handling low-frequency but semantically critical tokens. Our analysis suggests that mixture-of-experts style routing may be unnecessary for rare token processing, as universal connectivity already enables effective integration of specialized neurons.

Despite these insights, our study has several limitations. First, our findings are correlational: we observe robust patterns of plateau neuron recruitment and distributed access, but we do not establish causal mechanisms or developmental trajectories during training. Second, our analysis focuses on GPT-2 and Pythia families and specific network components, namely the final MLP layer and attention heads near the output. Whether these distributed mechanisms generalize across architectures, model scales, or modalities remains an open question. Third, ablation methods measure necessity but may overlook distributed redundancy, and token-matching controls cannot fully capture semantic complexity.

Future work should examine how plateau neurons emerge during training, extend analyses to diverse architectures, and employ causal interventions to test their functional necessity. By revealing how rare token specialization arises from distributed differentiation rather than modularity, our findings provide a foundation for understanding transformer interpretability and for guiding principled architectural design.

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

## A Detailed Methodological Procedures

### A.1 Graph Construction for Spatial Organization Analysis

We construct weighted, undirected graphs from neural activation data to assess functional connectivity patterns. For a given set of neurons, we compute edge weights using the Pearson correlation coefficient between activation vectors of neurons $i$ and $j$ across our corpus of 1,000 contexts:

$$w_{ij} = \text{corr}(\mathbf{a}_i, \mathbf{a}_j) \tag{5}$$

where $\mathbf{a}_i$ represents the activation vector for neuron $i$ across all contexts. The resulting graphs are signed, capturing both positive and negative correlations. We apply a threshold of $|w_{ij}| > 0.1$ to remove weak connections and focus on meaningful functional relationships.

### A.2 Community Detection Algorithms

**Louvain Algorithm:** Our primary community detection method uses the Louvain algorithm [2], which optimizes modularity through iterative local optimization and community aggregation. We run the algorithm 100 times with different random seeds and select the partition with highest modularity score to ensure robustness.

**Spectral Clustering:** As a validation approach, we employ spectral clustering on the graph's normalized Laplacian matrix. We perform eigendecomposition and embed nodes in a low-dimensional space (k=2 to k=8 communities) where clusters are identified via k-means clustering. This method is particularly effective for detecting complex community structures in signed graphs.

### A.3 Modularity Measures

We employ both standard and signed modularity measures to quantify clustering quality:

**Standard Modularity:**
$$Q = \frac{1}{2m} \sum_{ij} \left[ A_{ij} - \frac{k_i k_j}{2m} \right] \delta(c_i, c_j) \tag{6}$$

**Signed Modularity:** For signed networks, we use the extension:

$$Q_{\text{signed}} = \frac{1}{2m^+} \sum_{ij} \left[ A_{ij}^+ - \frac{k_i^+ k_j^+}{2m^+} \right] \delta(c_i, c_j) - \frac{1}{2m^-} \sum_{ij} \left[ A_{ij}^- - \frac{k_i^- k_j^-}{2m^-} \right] \delta(c_i, c_j) \tag{7}$$

where $A^+$ and $A^-$ represent positive and negative edge subgraphs respectively.

### A.4 Statistical Validation Procedures

**Control Group Generation:** For each experimental group (plateau neurons), we generate 100 random control groups of the same size from the full neuron population. This controls for baseline connectivity patterns and group size effects.

**Significance Testing:** We use Mann-Whitney U tests to compare modularity distributions between experimental and control groups, with Bonferroni correction for multiple comparisons. Effect sizes are calculated using Cohen's d.

**Bootstrap Confidence Intervals:** Modularity scores are estimated with 95

### A.5 Attention Head Ablation Implementation

**Ablation Procedure:** For each attention head $h$ in layers 20-30, we zero the attention weight matrix $\mathbf{W}_h$ and forward-propagate to measure downstream effects on plateau neuron activations. This is implemented through:

$$\text{Attention}_{\text{ablated}} = \text{softmax}\left(\frac{\mathbf{Q}\mathbf{K}^T}{\sqrt{d_k}} \odot \mathbf{M}_h\right)\mathbf{V} \tag{8}$$

where $\mathbf{M}_h$ is a binary mask with zeros for the ablated head.

**Control Ablations:** We perform control ablations on randomly selected heads and non-attention components (layer norms, feedforward weights) to establish baseline impact levels and ensure observed effects are attention-specific.

**Activation Patching:** For validation, we implement clean/corrupted activation patching [**?** ] where we replace attention outputs with activations from different input contexts to isolate causal contributions.

## A.6 Token Selection and Matching Criteria

**Frequency Thresholds:** Rare tokens defined as appearing $< 100$ times in OpenWebText training corpus; common tokens appearing $> 10,000$ times, based on GPT-2 tokenizer frequency distributions.

**Matching Procedure:** Each rare token is matched to a common token controlling for: (1) token length ($\pm 1$ character), (2) part-of-speech tag (using spaCy), (3) sentence position (beginning/middle/end), and (4) syntactic role when possible. Matching accuracy verified through linguistic feature analysis.

**Context Selection:** For each token pair, we sample 20 diverse contexts ensuring grammatical validity and semantic coherence. Contexts are drawn from different domains (news, literature, technical writing) to test generalization across linguistic environments.

