# OpenReview forum: "No Clustering, No Routing: How Transformers Actually Process Rare Tokens"
_NeurIPS.cc/2025/Workshop/UniReps — UniReps2025_

### Official Review · Reviewer_1ihJ · 2025-09-06
**A Review of Rare Token Processing in Transformers**

**Confidence:** 3

**Review:**

### **Summary**
The paper explores the question of how large language models process rare tokens from the viewpoint of mechanistic interpretability. The authors conduct their study on GPT-2 and Pythia models, arguing that rare tokens specialization arises through distributed, training-driven specialization rather than architectural modules.

### **Strengths**
- The paper does a good job of framing a clear motivation of pursuing the understanding how LLMs represent long-tail / rare tokens.
- Their multi-pronged analysis (neuron influence, spatial organization, attention routing) is a strong design choice, as multiple methodologies converge on a similar conclusion.
- The insight is quite interesting and provides a counterpoint to the mixture-of-experts literature where specialization occurs through architectural modularity, while the authors show that specialization can be implemented implicitly through distribution representations.

### **Weaknesses**
- There is an inconsistent definition of rare tokens. Section 2 uses a 50th-percentile split, while Appendix A.6 uses fixed frequency cutoffs (<100 vs >10,000).
- The paper has a significant number of null findings (Table 1: no significant clustering, Table 2: minimal impact on plateau neuron activation). Although this is across 2 models and offer important observations, the outcome might be an artifact of the specific algorithm used or choice of correlation metric. There is also a lack of causal findings since the study relies more on correlation. Yet, the authors make strong claims.

### **Questions**
- Can you clarify how are rare and common tokens decided? In line 46, it is mentioned that 50th percentile is taken as a threshold while Appendix A.6 defines frequency thresholds.
- How robust are the results to different cutoffs (δ threshold, correlation threshold, clustering parameters)?

### **Conclusion**
The study is limited to only the final MLP. While the paper discusses this as a limitation, a further analysis into intermediate layers as well as models can lead to interesting insights. Does this distributed specialization hold for early layer? How does the specialization vary as you go deeper into the model? What would be the interplay between architectural modularity (e.g. mixture-of-experts) and distribution specialization? All these are directions which has potential.

**Score:**

3

**Topic Fit:**

2

---

### Official Review · Reviewer_B1KV · 2025-09-15
**Solid evidence for distributed rare token processing in transformers, though scope remains limited**

**Confidence:** 3

**Review:**

Strengths:
- The research digs into how transformer models actually process information internally - particularly the tricky question of whether they organize specialized functions into separate modules or spread them throughout the network.
- Didn't rely on just one experimental approach. They measured individual neuron contributions, mapped the spatial arrangement of specialized neurons using network analysis techniques, and examined attention patterns
- Included important reality checks. Compared their results against random chance, applied statistical tests to verify their findings weren't flukes
- Instead of drawing conclusions from a single model, they tested their ideas on both GPT-2 XL and Pythia architectures
- Systematically examines how specialized neurons are physically distributed within transformer networks
- Found novel ways to apply social network analysis tools to understand neural organization
- The paper establishes precise criteria for identifying "plateau neurons" - the specialized units that handle rare tokens differently than common ones

Research Methods
- Measured how much each neuron mattered by systematically disabling individual neurons and observing the performance changes. Paired this with proper statistical controls to ensure reliable results.
- Using two different clustering algorithms (Louvain and spectral methods), they mapped out whether specialized neurons formed discrete groups or remained scattered throughout the network layer
- The study combined correlation analysis with targeted experiments where specific attention components were removed to understand how information flows to specialized neurons
- Included multiple comparison corrections, effect size calculations, confidence intervals, and a substantial sample of over 25,000 tokens from a standard dataset

Significance:
- The findings help resolve whether transformer models use modular organization or distributed processing
- The results suggest that existing transformer architectures may handle rare tokens effectively without needing specialized routing systems like mixture-of-experts models that some researchers have proposed

Weaknesses:
- The investigation focuses only on the final processing layer and late-stage attention mechanisms. Earlier layers might show completely different organizational patterns.
- Testing occurred on just two model families. Different architectures, larger models, or systems trained on different data might organize processing differently.
-While the study documents clear patterns in neural behavior, it doesn't prove these patterns are necessary for rare token processing. The research shows what happens but not definitively why it must happen this way.
-Don't explain how these specialized neurons develop during model training, whether the observed patterns are essential features or coincidental byproducts, or what mechanisms drive the dual processing systems they discovered.

Overall Assessment:
The experimental design is rigorous, the findings are clear and well-supported, and the theoretical implications are significant. While limited in scope and primarily correlational, it provides valuable insights into how transformers achieve specialization through distributed rather than modular mechanisms.

**Score:**

4

**Topic Fit:**

3

---

### Official Review · Reviewer_KymG · 2025-09-15

**Confidence:** 3

**Review:**

**Summary**

This paper investigates how transformer-based language models process rare tokens, focusing on plateau neurons -- a small subset of MLP neurons in the final transformer layer that strongly influence rare token prediction. Based on the community detection analysis of a co-activation graph and attention heads ablation on GPT-2 XL and Pythia models, authors show that these neurons are spatially distributed within the last MLP layer rather than forming modular clusters, and they are accessed via general attention patterns, with no evidence of selective routing.

**Strengths**
1. The paper builds upon prior work [1], which identified plateau neurons involved in rare token processing but didn’t investigate their structural organization. This study provides additional insights into their behavior, advancing our understanding of the inner workings of language models.
2. The use of statistical testing to validate the significance of results is noteworthy. While such rigor should be standard in ML research, it is often overlooked, and this paper sets a positive example.
3. The paper is generally well-structured and clearly written, making the methods and findings accessible, with only a few minor exceptions noted below.


**Weaknesses**
1. In lines 100 and 107, the authors claim that “rare tokens recruit plateau neurons not activated by common tokens.” Based on the provided experiments, I’m not convinced think this conclusion is fully supported. The results suggest that plateau neurons are more influential for rare tokens, but this doesn’t necessarily imply that common tokens don’t activate them at all.
2. Prior work, e.g. [2, 3] has shown that attention heads can often be dynamically pruned during inference with little to no impact on model performance. Given this, it is perhaps unsurprising that ablation of attention heads might have no significant effect on the activations of plateau neurons—or of any other neuron subset.
3. As acknowledged in the limitations section, the analysis is restricted only to 2 relatively old models – Pythia 410M and GPT-2 XL (1.5B). Extending the study to more recent models (e.g., Llama or Gemma families) would strengthen the contribution. Moreover, investigating instruction-tuned or reasoning models would provide valuable insights into how different post-training strategies affect the processing of rare tokens.


**Minor comments and questions**
* In line 49, what does “mean ablation experiment” refer to?
* In Equation 1, what exactly is $\mathcal{L}_{\text{baseline}}$? Is it the loss without any neuron intervention?
* Does Figure 1 present results for Pythia, GPT-2, or both models?
* The axes in Figure 1 would benefit from more informative labeling.
* In lines 55–57, a reference to Appendix A1 explaining the construction of the co-activation graph would be helpful.
* What are the ‘excitatory and inhibitory plateau neurons’ (line 82 and Table 1)? I couldn’t find their definitions in the paper.
* In Equation 3, what does “Activation” denote? Additionally, where in the paper is this measure later used?

**References**

[1] Liu, Jing, Haozheng Wang, and Yueheng Li. "Emergent Specialization: Rare Token Neurons in Language Models." arXiv preprint arXiv:2505.12822 (2025).

[2] Michel, Paul, Omer Levy, and Graham Neubig. "Are sixteen heads really better than one?." Advances in neural information processing systems 32 (2019).

[3] Liu, Zichang, et al. "Deja vu: Contextual sparsity for efficient llms at inference time." International Conference on Machine Learning. PMLR, 2023.

**Score:**

4

**Topic Fit:**

3